# In Situ Modification of Multi-Walled Carbon Nanotubes with Polythiophene-Based Conjugated Polymer for Information Storage

**DOI:** 10.3390/ma16030908

**Published:** 2023-01-18

**Authors:** Wei Li, Xiaoyang Wang, Bin Zhang, Yu Chen

**Affiliations:** 1Key Laboratory for Advanced Materials, Joint International Research Laboratory of Precision Chemistry and Molecular Engineering, School of Chemistry and Molecular Engineering, East China University of Science and Technology, Shanghai 200237, China; 2Guangxi Key Laboratory of Information Material, Engineering Research Center of Electronic Information Materials and Devices, School of Material Science and Engineering, Guilin University of Electronic Technology, Guilin 541004, China

**Keywords:** carbon nanotubes, polymer memory, surface modification, information storage

## Abstract

One-dimensional multi-walled carbon nanotubes (MWNTs) have unique electrical properties, but they are not solution-processable, which severely limits their applications in microelectronic devices. Therefore, it is of great significance to improve the solubility of MWNTs and endow them with new functions by chemical modification. In this work, MWNTs were in situ functionalized with poly[(1,4-diethynyl-benzene)-*alt*-(3-hexylthiophene)] (PDHT) via Sonogashira–Hagihara polymerization. The obtained material PDHT-*g*-MWNTs was soluble in conventional organic solvents. By sandwiching a PDHT-*g*-MWNTs film between Al and ITO electrodes, the fabricated Al/PDHT-*g*-MWNTs/ITO electronic device exhibited nonvolatile rewritable memory behavior, with highly symmetrical turn-on/off voltages, a retention time of over 10^4^ s, and durability for 200 switching cycles. These findings provide important insights into the development of carbon nanotube-based materials for information storage.

## 1. Introduction

Since their discovery in 1991 by S. Iijima [1], carbon nanotubes (CNTs) have made significant breakthroughs in many new fields such as nanoelectronics, super-strong composites, hydrogen storage materials, and catalyst carriers because of their exclusive structural, electrical, and mechanical characteristics [2,3]. Their superior performances have attracted interest from the physical, chemical, and materials science communities worldwide [4,5,6,7]. However, their low solubility in common organic solvents has greatly restricted their applications [8,9,10]. Due to continuous improvements in CNT preparation methods, chemical modification of CNTs has been used to functionalize them to broaden their application scope. Atomic force microscopy (AFM) observations have shown that in more dilute solutions of <5 mg/L, CNTs may exist as individual nanotubes, while in more concentrated solutions, they are mainly found as ropes or bunches because of van der Waals interactions inside them [11]. There have been numerous reports of functionalization reactions of SWNTs and MWNTs [12,13,14], which can be broadly divided into covalent and noncovalent modification ways. The covalent attachment of functional groups to CNTs is an effective way of modifying them to improve their solubility, increase their dispersion, and create chemical bonds between their surface and polymers [15].

Conjugated polymers often have special photoelectric properties, such as light absorption and fluorescence emission, due to their conjugated structures [16,17,18,19,20]. In addition, conjugated polymers have super extensive applications for the preparation of low-cost, stretchable electronic devices including organic field-effect transistors (OFETs), light-emitting diodes (LEDs), and organic photovoltaic (OPV) cells [21,22,23]. Amongst the various conductive polymers, polythiophene and polythiophene-based polymers are important due to their good processability and stability [24,25,26]. Both CNTs and conjugated polymers have electronically conjugated structures, and each have their own unique optoelectronic and chemical properties. Therefore, the covalent bonding of polythiophene derivatives to carbon nanomaterials can improve their electrical conductivity while increasing the solubility of carbon nanotubes [27]. These characteristics have opened opportunities for the practical application of highly stable nonvolatile resistive random access memory (RRAM) derived from graphene nanomaterials.

Although there have been many reports on memory devices based on graphene-like low-dimensional nanomaterials [28,29,30,31], conjugated polymer-functionalized carbon nanotube materials for nonvolatile RRAMs have been rarely reported. In this work, through utilizing 4-bromobenzene modified MWNTs (MWNTBr) as a one-dimensional template, the in situ growth of π-conjugated polythiophene derivatives from MWNTs surfaces was realized via the Sonogashira–Hagihara reaction. The generated material, PDHT-*g*-MWNTs, was used for memory device applications. The manufactured device with the Al/PDHT-*g*-MWNTs/ITO structure displayed typical nonvolatile bistable resistive switching behavior with +2.35 V (writing) and −2.45 V (erasing) voltages, high ON/OFF ratios (>10^3^), long retention times, outstanding durability, extremely symmetrical writing/erasing voltages, and repetitive “write-read-erase-read-write” capability. As shown in Table 1, our synthesized material had a lower |ΔON-OFF| value than similar materials in other reported work, and the storage performance of the device remained virtually unchanged based on the retention test results, indicating its potential application in high-stability data storage.

## 2. Materials and Methods

### 2.1. Measurements and Instrument

All reagents and solvents were used as received without further purification. Multi-walled carbon nanotubes without functional groups on the surface were purchased from Macklin. Their purity, average inner diameter, average outer diameter, and aspect ratio were ≥95%, 8 nm, 24 nm, and 415–1250, respectively. ^1^H NMR spectra were recorded on a Bruker AVANCE 400 NMR spectrometer, operating at frequencies of 400 MHz using CDCl_3_ as solvent. Molecular weights were obtained on a Waters 2690 gel permeation chromatography (GPC). High magnification transmission electron microscopy (TEM) was performed using a JEOL-2100 (JEOL Ltd., Tokyo, Japan) TEM system operating at an accelerating voltage of 200 kV. Raman spectra were recorded on an Invia/Reflrx Laser Micro-Raman spectrometer (Renishaw, England) with an excitation laser beam wavelength of 514 nm. Thermal gravimetric analyses (TGA) were performed using Pyris 1 TGA in a nitrogen atmosphere. Atomic force microscopy (AFM) measurements were performed on a Solver P47-PRO (NT-MDT Co., Moscow, Russia) microscope. X-ray photoelectron spectroscopy (XPS) experiments were carried out on a Kratos AXIS Ultra DLD system incorporating a 165 mm hemispherical electron energy analyzer, which used a monochromatic Al Kα source (hν = 1486.6 eV) operating at 150 W. Survey scans (0–1200 eV) were performed to identify constitutive elements. Survey wide-scan spectra (160 eV pass energy and 1.0 eV step size) and high-resolution spectra of C 1s, S 2p lines (20 eV pass energy and 0.05 eV step size) were collected. The base pressure in the analysis chamber was 1.0 × 10^−9^ torr and during sample analysis 1.0 × 10^−8^ torr. All narrow-scan spectra were corrected for the background using the Shirley approach. EPR experiments were performed using a Bruker EMX-Nano Benchtop spectrometer.

### 2.2. Synthesis of PDHT

Anhydrous DMF (25 mL) was heated to 80 °C while a combination of 130 mg 2,5-dibromo-3-hexylthiophene (0.40 mmol), 57 mg 1,4-diethylbenzene (57 mg, 0.45 mmol), 35 mg Pd(PPh_3_)_4_ (0.03 mmol), 7 mg CuI (0.04 mmol), and Et_3_N (2 mL) was added. After 72 h, methanol solution was added to the reaction system. Then, the precipitate is formed and filtered. The monomers and residual salts were removed by washing the acquired precipitate with methanol and deionized water. An overnight vacuum drying process at 60 °C yielded 147 mg of a brown-yellow powder. The *M*_n_ of this polymer was 1.15 × 10^4^. ^1^H NMR (400 Hz, CDCl_3_): 7.6–7.4 (m, 6H), 3.30–3.27 (t, 4H), 1.9 (t, 4H), 1.7–1.6 (m, 4H), 1.2–1.0 (m, 8H), and 0.6 (m, 4H).

### 2.3. Synthesis of MWNTBr

In total, 1000 mg 4-Bromobenzenediazonium tetrafluoroborate in water was poured into a dispersion of 100 mg MWNTs in 50 mL dry acetone while being stirred magnetically at room temperature. The obtained product was filtered and thoroughly cleaned with a lot of common solvents after 2 h. A total of 107 mg of MWNTBr were produced after drying under a vacuum for 24 h at 60 °C.

### 2.4. Synthesis of PDHT-g-MWNTs

In total, 80 mg of MWNTBr was added to 50 mL of anhydrous DMF under the protection of argon, and then sonicated for 30 min at room temperature to obtain the DMF dispersion of MWNTBr. The obtained dispersion was infused with a combination of 811 mg 2,5-dibromo-3-hexylthiophene (1 mmol), 139 mg 1,4-diethynylbenzene (1.1 mmol), 35 mg Pd(PPh_3_)_4_ (0.03 mmol), 7 mg CuI (0.04 mmol), and 4 mL Et_3_N. The reaction mixture was added to methanol after stirring for 72 h at 80 °C and then vacuum-filtered by a polycarbonate film (ϕ 0.22 μm). The collected filter cake was thoroughly rinsed with methanol and deionized water until the filtrate was clear. In total, 303 mg of PDHT-*g*-MWNTs was produced after vacuum drying at 60 °C overnight.

### 2.5. Device Preparation and Characterization

The ITO glass substrate underwent 15 min of meticulous pre-cleaning in an ultrasonic bath. On the ITO glass substrate, 100 μL of PDHT-*g*-MWNTs in NMP (5 mg/mL) was spin-coated for 10 s at 700 rpm, followed by 45 s at 1400 rpm. The ITO glass substrate was then removed while being vacuum dried. Aluminum electrodes were evaporated on the surface of the PDHT-*g*-MWNTs film using the vacuum coating instrument at 10^−7^ Torr. Each top electrode was 500 μm in diameter and about 200 nm in thickness. The contact area of aluminum electrodes was approximately 0.785 mm^2^. All performance tests were conducted on a Keithley 4200 without device packaging and under normal temperature and pressure.

## 3. Results and Discussion

As shown in Figure 1, 4-bromobenzene-functionalized MWNTs (MWNTBr) were first obtained through the reaction of MWNTs and 4-bromo-benzenediazonium tetrafluoroborate. Then, by using MWNTBr as a one-dimensional template, a novel polythiophene-based conjugated polymer was grafted from the MWNTs’ surface via Sonogashira–Hagihara polymerization to produce PDHT-*g*-MWNTs (Appendix A). Transmission electron microscopy was used to examine the MWNTs both before and after modification with conjugated polymers in order to comprehend the surface properties of the materials. As shown in Figure 1a,b, the MWNTs were interspersed with a uniform wall surface and an average diameter of approximately 24 nm. After in situ modification with the conjugated polymer PDHT, the wall surface became significantly rougher. A layer of material was attached to the wall surface, and the average diameter increased by approximately 39 nm. This indicated that the conjugated polymers were grafted onto the surface of the tube and formed a film with an average thickness of approximately 19.5 nm, which covered the CNT.

Infrared spectroscopy confirmed the presence of PDHT in PDHT-*g*-MWNTs (Figure 1c). MWNTBr displayed several characteristic peaks at 1627, 1458, 1122, 1066, and 613 cm^−1^. The peaks at 1627 and 1458 cm^−1^ were attributed to the C=C stretching of graphitic domains. The stretching vibration peak at 1066 cm^−1^ was attributed to C-Br, indicating the successful introduction of bromine on the surface of MWNTs. In the PDHT spectrum, the characteristic peak of C≡C at 2193 cm^−1^ disappeared. Peaks also emerged at the same locations in the spectra, suggesting that the covalent functionalization of MWNTs with PDHT may have occurred through nitrite chemistry. The Raman spectra of MWNTs (Figure 1d) with 514 nm laser excitation revealed a ratio (*I*_D_/*I*_G_) of 1.01, which had double different bands at 1593 (G-band) and 1309 (D-band) cm^−1^. The G-band red-shifted to 1603 cm^−1^ after covalently grafting bromophenyl groups onto the MWNTs surface. The increase in the relative intensity of the D-band relative to the G-band had been used as an indication of sidewall covalent functionalization, as it reflected the transition from *sp*^2^ to *sp*^3^ hybridization of carbon atoms on the nanotube wall [36]. The MWNTBr’s *I*_D_/*I*_G_ increased to 1.16 relative to MWNTs’, but the *I*_D_/*I*_G_ decreased to 0.79 after the in situ grafting of PDHT. In the case of PDHT-*g*-MWNTs, although some *sp*^2^ carbons in the MWNTs had been converted to *sp*^3^ carbons, they were still less than the *sp*^2^ aromatic carbons in the polymer chain. Furthermore, the peak belonging to the alkyne group at 2190 cm^−1^ in PDHT was also present in PDHT-*g*-MWNTs. These results showed that the designed polymer was covalently grafted to MWNTs.

To determine the successful functionalization of MWNTs and estimate their content in the final product, the TGA used nitrogen protection and a temperature range of 35 to 800 °C. As evident from the spectrum (Figure 1e), MWNTBr was more thermally unstable than MWNTs. When heated to 143 °C, MWNTBr began to lose weight because of the decomposition of bromine-containing groups on the MWNTs’ surface. The sudden increase in the weight loss rate at temperatures up to 541 °C was attributed to the onset of decomposition of the main body of the MWNTs. In marked contrast to MWNTBr, MWNTs had a much slower and more stable TGA curve, which is evidence of the successful functionalization of MWNTs. MWNTs had a weight loss of 5.03% in the range of 35–800 °C. If the weight loss ratio of MWNTs in MWNTBr was consistent with that of MWNTs, then the content of MWNTs in MWNTBr can be estimated to be 92.90% (87.87% + 5.03%). PDHT exhibited two weight loss intervals in the range of 149–300 °C (3.04%) and 300–800 °C (42.17%). After modification, the PDHT-*g*-MWNTs curve showed less weight loss than that of PDHT, which suffered 1.23% and 25.20% weight loss in the region of 149–300 °C and 300–800 °C, separately. By examining the temperature range of 35–800 °C, the total weight loss of MWNTs, PDHT, and PDHT-*g*-MWNTs was 5.03%, 45.65%, and 26.43%. By supposing that the PDHT remnant of PDHT-*g*-MWNTs at 800 °C had an equal weight percentage as that of PDHT, the content of MWNTs in PDHT-*g*-MWNTs was estimated at 24.25% (5.03% + 45.65% − 26.43%).

Two indications of carbon/bromine elements were visible in the MWNTBr’s wide-scan XPS spectrum (Figure 1f) at binding energies of 284 eV (C 1s), 256 eV (Br 3s), 183 eV (Br 3p), and 70 eV (Br 3d). The peaks of carbon emerged at 284.6 eV (C=C), 285.4 eV (C-C), and 286.3 eV (C-Br), in accordance with the MWNTBr C 1s core-level XPS spectrum (Figure 1g). The successful modification of bromobenzene on the surface of carbon tubes was illustrated by the XPS spectrum combined with previous IR and Raman spectra. The peaks at 285.8, 284.1, 285.2, and 284.6 eV corresponded to C-S, C≡C, C-C, and C=C in the PDHT-*g*-MWNTs C 1s core-level XPS spectrum (Figure 1h). In the S 2p core-level XPS spectrum of PFTC-RGO (Figure 1i), the peaks at 164.0 eV and 165.1 eV belonged to S 2p_3/2_ and S 2p_1/2_, respectively. These results proved that the PDHT was grafted onto the MWNTs surface.

The UV-vis absorption spectra of MWNTs, PDHT, and PDHT-*g*-MWNTs was illustrated in Figure 2a. The MWNTs showed no significant peaks in the region of 310–600 nm, showing a wide absorption that diminished at higher wavelengths. The PDHT spectrum contained a maximum absorption peak at 397 nm. In the PDHT-*g*-MWNTs spectrum, the peak at 406 nm underwent a red-shift of 9 nm relative to the peak of PDHT, indicating possible electronic interactions between MWNTs and PDHT. We used steady-state fluorescence measurements and electron paramagnetic resonance (EPR) technology to investigate the photoinduced intramolecular events that happened in PFTC-RGO, which included PDHT donor and MWNTs acceptor units. The fluorescence spectrum of PDHT-*g*-MWNTs in toluene contained a significant emission peak at 562 nm, as shown in Figure 2b. The strong emission peak red-shifted from 562 nm in toluene to 568 nm in tetrahydrofuran (THF) to 573 nm in *N*-methyl-pyrrolidone (NMP) as the polarity of the organic solvents enhanced. This was followed by a progressive drop in the fluorescence strength and a gradual broadening of the peak. These results showed that electron transfer from PDHT to MWNTs* may have been responsible for the fluorescence quenching process. When one species was photoexcited, photoinduced electron transfer occurred. The radical cation of the donor and the radical anion of the acceptor are produced as a result of an electron being transferred from an electron-donating species to an electron-accepting species in this simple process [37]. The EPR spectra of PDHT-*g*-MWNTs exhibited the characteristic radical signal in Figure 2c before illumination, showing the presence of free radicals or unpaired electrons in the PDHT-*g*-MWNTs system. The signal strength significantly decreased after irradiation from a 450 nm beam for approximately half a minute under environmental lighting. The primary cause of this phenomenon was the PDHT-covalently grafted MWNTs system’s photoinduced intramolecular electron transfer effect.

The beginning oxidation/reduction potentials (*E*_ox_/*E*_red_) were located at +0.77/−0.73 V versus Ag/Ag^+^, which corresponded to +1.06/−0.44 V vs. saturated calomel electrode (SCE), from the cyclic voltammogram curve of the PDHT-*g*-MWNTs film in Figure 2d. The electron affinity (EA), the ionization potential (IP) and the HOMO/LUMO energy levels of PDHT-*g*-MWNTs were determined by the equations in the literature [32]. The bandgap values of PDHT-*g*-MWNTs, EA, IP, LUMO, and HOMO were 1.50, 3.95, 5.45, −3.98, and −5.48 eV, respectively. 

As shown in Figure 3a, the MWNTs were almost uniformly dispersed in the solution, while PDHT (Appendix A) and PDHT-*g*-MWNTs dissolved completely. After placing the two bottles of solution for 24 h (Figure 3b), MWNTs were suspended, and sedimentation was distributed at the upper and lower layers of the solution. The solution of PDHT-*g*-MWNTs did not undergo any change visible to the naked eye. The digital photographs showed that the in situ grafting of polythiophene-based conjugated macromolecules on the surface of MWNTs greatly improved the solubility of MWNTs in NMP. The memory device configuration was shown in Figure 3c. The active layer was observed by atomic force microscopy (AFM) and was placed in the upper-right corner of the device schematic. The surface roughness of the AFM image of the PDHT-*g*-MWNTs film was 1.81 nm, indicating that the material formed a relatively homogeneous film on the substrate due to the good solubility of MWNTs modified by PDHT in organic solvents. The conductivity of the as-fabricated device quickly shifted from the OFF state (10^−5^ A) to the ON state (10^−2^ A) at the threshold voltage of +2.35 V, as displayed in Figure 3d, suggesting the occurrence of a writing process. During the following forward sweep, the device was maintained in the ON state. Although the electric source was switched off, the ON state remained constant. This trait suggested that this device’s memory performance was non-volatile. That was to say, the memory system of the computer could store vital data without an electric source. The ON state reverted to its initial OFF state while a negative sweep voltage from 0 to −3 V was added to the device, as evidenced by the quick drop in current from 10^−2^ to 10^−5^ A at the threshold voltage of −2.45 V. This procedure corresponds to the erasure procedure. The device remained in a high resistance state for the remainder of the following backward sweep. The OFF state was then reprogrammed to the ON state after a positive sweep was added. These findings demonstrate that with an ON/OFF current ratio of >10^3^, the as-designed novel electronic device may function as non-volatile rewritable memory.

Replacing existing silicon-based materials with polymer-modified carbon materials for information storage requires symmetrical values of turn-on and turn-off voltages, a high ratio of OFF to ON current, repeatability, and device stability. After measuring the threshold voltage for 80 write-erase cycles (Figure 3e), the distribution of the threshold voltage was very concentrated, and the absolute difference between the on and off voltage was only 0.1 V. This indicates that the device had good fault tolerance and could be used with a high degree of confidence. At a read voltage of +2.35 V, a high ON/OFF current ratio of >10^3^ was achieved, as illustrated in Figure 3f. After analyzing the switching ratio data, a Gaussian fit was performed. The fitted curve reflected the distribution of the data was intensive, and the switching ratio was stable. The Weibull distribution (Figure 3g) shows that the distribution of resistances at +1.0 V in the high conductivity and low conductivity states was quite concentrated. Figure 3h shows the stability of the device at +1.0 V in both the ON and OFF states. According to the contact area of the aluminum electrode, the current density through the composite in the ON and OFF states at +1.0 V could be estimated as 2.5 × 10^−2^ A/mm^2^ and 5.3 × 10^−6^ A/mm^2^, respectively. Throughout the measuring process, for more than 10,000 s, there was no appreciable deterioration in the current for the ON or OFF conditions. This outcome indicated that the memory device had strong stability and an imperceptible rate of misreading. The dependability and stability of flash-type memory devices are correlated with their retention capacity and read cycle count. Cycling endurance tests for the rewritable PDHT-*g*-MWNTs memory device (Figure 3i) were performed to show the dependability of the memory device. The first switching cycle involved a hold for 10 at +3 V, a pause at 0 V for 10 μs, then a switch to −3 V for 10 μs, and a final pause at 0 V for 10 μs. The resistance in the ON and OFF states remained stable even after 200 cycles of switching. A set of programs that could accurately mimic the environment of the device without outside power were developed to demonstrate the stability of the ON and OFF states after removing the external electric source. These programs used very short pulse times (20 μs) and very small pulse voltages (+0.1 V). As shown in Figure 3j, the device maintained its ON/OFF states without deterioration over the test time of 1.0 × 10^5^ s after being switched between the ON and OFF states under an externally applied voltage of +0.1 V.

In the case of Al/PDHT-*g*-MWNTs/ITO, the energy barrier (0.30 eV) between the LUMO energy of PDHT-*g*-MWNTs (−3.98 eV) and the work function of Al (−4. 28 eV) was smaller than that (0.68 eV) between the HOMO energy of PDHT-*g*-MWNTs (−5.48 eV) and the work function of ITO (−4.80 eV), indicating that the injection of electrons from the Al electrode into the LUMO of PDHT-*g*-MWNTs was a favorable process (Figure 4a). Similar to C_60_ and graphene, the large number of aromatic rings made carbon nanotubes a good electron acceptor (A), while thiophene was a good electron donor (D) and hole transporter. For the D-A system, both inter- and intra-molecular charge transfer occurred readily. In the absence of input voltage, the electrons were stable, so the fabricated Al/PDHT-*g*-MWNTs/ITO device was in a low conductivity (OFF) state. As shown in Figure 4b, the electrons located in the HOMO−1 easily transited to LUMO when the input voltage was applied. However, the transition of electrons from HOMO to LUMO was limited due to the lack of overlap between HOMO and LUMO. In addition, the electrons in HOMO could spontaneously transfer to HOMO−1 and filled the holes created in HOMO−1. Due to the huge conjugation system in MWNTs, the electrons in LUMO were transferred to LUMO+1. Thus, a conductive charge transfer state was generated and could be efficiently stabilized by the conjugation system in MWNTs, which led to the nonvolatile nature of the prepared memory devices. The charge transfer path formed along with the increase in charge transfer interactions would turn the device on (Figure 4c). When an opposite forward pressure was applied, the electrons located in the MWNTs could be extracted and therefore the device could be returned to the OFF state. As shown in Appendix A, the OFF state current of the device can be fitted by a space-charge-limited current (SCLC) model, while the ON state current can be fitted by an Ohmic current model.

## 4. Conclusions

In summary, PDHT-g-MWNTs, soluble polythiophene-based conjugated polymer-functionalized MWNTs, were in situ synthesized via Sonogashira–Hagihara polymerization. The as-fabricated Al/PDHT-*g*-MWNTs/ITO electronic device displayed a nonvolatile rewritable memory effect, extremely symmetrical writing and erasing voltages, durability for more than 200 switching cycles, and a retention duration of more than 10^4^ s. These results showed that PDHT-*g*-MWNTs were a promising material for constructing excellent-performance information storage devices.

## Data Availability

Not applicable.

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
