# Peer review of "In Situ Modification of Multi-Walled Carbon Nanotubes with Polythiophene-Based Conjugated Polymer for Information Storage"

_materials, 2023, doi:10.3390/ma16030908_

Round 1
Reviewer 1 Report
The work is devoted to an extremely topical direction in the development of electronics based on carbon structures. The authors claim that they have succeeded in fabricating an RRAM device based on MWCNTs functionalized with PDHT. The results of testing the received device are quite impressive, but there are a few notes that are worth working on.
The experimental part is not described in detail, with large gaps:
1) What MWCNTs were used in the work? Their purity, average diameter, aspect ratio, functional groups on the surface?
2) What was used to obtain the Raman spectra?
3) Section 2.4, how was the sonication of MWCNTs? This is important for understanding the presence of MWCNT agglomerates in the future.
4) Characteristics of atomic force microscopy should be given
5) Contact area of aluminum electrodes? Current density through the composite?
The authors write: The G band could show the extent of MWNTs alteration, and the D band could reflect the covalent functionalization process that changed from sp2 to sp3 hybridization. This is incorrect, you must also use the appropriate reference
The XPS technique is not described in sufficient detail, and the binding energies used to interpret the spectra lead to a conclusion about the low reliability of the XPS data obtained. sp1 283.8-284.1 eV, sp2 (С=С) 284.6 eV, sp3 (С-С) 285.2-285.4 eV Did the authors use subtraction of the Shirley-type background to correct the obtained data?
What is the reason for the sharp and reproducible change in the conductivity of the obtained composites? Formation of new contacts between MWNT? Changing the state of the polymer on the surface? The authors publish a second article on a similar topic, with a difference in the polymer used, but in both cases they confine themselves to presenting experimental data without attempting to make assumptions about the nature of the observed phenomena.
Reviewer 2 Report
Dear authors, the article In-situ Modification of Multi-walled Carbon Nanotubes with Polythiophene-1 based Conjugated Polymer for Information Storage is a very interesting study with new perspectives on applications of carbon nanotubes in the field of information storage. However, I would like to point out some points need to be clear, such as:
1. How the electrical conductivity of the carbon nanotubes change after their functionalization with PDHT? What is the difference of electrical conductivity between the MWNTs blank, the MWNTsBr and the final material PDHT-g-MWNTs? The functionalization affects the the electronic properties of MWNTs, as I suppose. However, independent of the result, the difference that occurs, is another point of their succesful functionalization of CNTs.
2. Concerning the UV-Vis of the PDHT-g- MWNTs seems to be totally different of the blank MWNTs. How does it explain that? Why so much excess of the polymer concentration during functionalization of CNTs?
3. The colour of the material PDHT-g- MWNTs seems to be yellow. What is the colour of the PDHT polymer without carbon nanotubes? How you can check the purification process of the PDHT-g-MWNTs?
4. What about the RAMAN spectra of PDHT-g-MWNTs? If ID/IG seems to be lower after the functionalization. How does it occur that?
5. It would be ideal, to give some more details of carbon nanotubes and theirs performance in the information storage field from older published works.
Round 2
Reviewer 1 Report
I think that the authors took into account all the previous comments
and the article should certainly be published in present form.